# Global Lessons from COVID-19: Regional Variations in the Management of Hospital-Acquired Infections During and Post-Pandemic

**DOI:** 10.3390/jcm14186654

**Published:** 2025-09-22

**Authors:** Corina Voinea, Elena Mocanu, Cristian Opariuc-Dan, Elena Dantes, Alexandra-Cristina Gache, Sorin Rugina

**Affiliations:** 1Medical Doctoral School, Faculty of Medicine, Ovidius University of Constanta, 1 University Alley, Campus—Corp B, 900470 Constanta, Romania; corina.badescu2012@gmail.com (C.V.); elena.dantes@365.univ-ovidius.ro (E.D.); alexandra.belu@365.univ-ovidius.ro (A.-C.G.); sorinrugina@yahoo.com (S.R.); 2Public Health Directorate Constanta, 1 Lacramioraei Alley, 900643 Constanta, Romania; 3Department of Public Health and Management, Faculty of Medicine, Ovidius University of Constanta, 1 University Alley, Campus—Corp B, 900470 Constanta, Romania; 4Department of Administrative Sciences, Faculty of Law and Administrative Sciences, Ovidius University of Constanta, 1 University Alley, Campus—Corp A, 900470 Constanta, Romania; cristian.opariuc@365.univ-ovidius.ro; 5Clinical Hospital of Pneumopthisiology Constanta, 40 Santinelei Street, 900002 Constanta, Romania; 6Department of Pneumology, Faculty of Medicine, Ovidius University of Constanta, 1 University Alley, Campus—Corp B, 900470 Constanta, Romania; 7Romanian Academy of Medical Sciences, 1 I.C. Brătianu Boulevard, Sector 3, 030171 Bucharest, Romania; 8Academy of Romanian Scientists, Ilfov Street, No.3, Sector 3, 050044 Bucharest, Romania

**Keywords:** healthcare-associated infections, COVID-19 pandemic, infection control measures, regional variation, global health strategies, hospital-acquired infections

## Abstract

**Background/Objectives**: The COVID-19 pandemic has significantly disrupted healthcare systems worldwide, exposing longstanding weaknesses, particularly in the prevention and control of healthcare-associated infections (HAIs). Regional disparities in infection prevention and control (IPC) strategies offered valuable lessons for improving public health preparedness. This systematic review aims to identify and compare regional IPC approaches adopted during and after the pandemic, highlighting best practices to strengthen healthcare resilience. **Methods**: The review was conducted in line with PRISMA guidelines and registered in the PROSPERO database (CRD420251032525). Articles published between 1 January 2020 and 31 March 2025, were retrieved from PubMed, Scopus, and Web of Science. Only full-text studies in English were included. The risk of bias was assessed using the ROBINS-I tool. **Results**: Of the 63 articles initially identified, 8 met the inclusion criteria. The selected studies demonstrated substantial variability in the implementation of IPC. The availability of infrastructure, funding, coordination capacity, and training of medical staff had a significant impact on outcomes. In regions with well-defined protocols and a solid infrastructure, there was a significant decrease in HAIs, while in resource-poor areas, there was a significant increase. Effective measures included continuous monitoring, regular staff training, provision of adequate equipment, expansion of testing capacity, reorganisation of hospitals, and introduction of technological innovations in healthcare. **Conclusions**: COVID-19 emphasised the importance of adaptable IPC frameworks. Strengthening health systems requires context-specific standards, sustained investment in infrastructure, continuous training, and increased international cooperation to better prepare for future health emergencies.

## 1. Introduction

Healthcare-associated infections (HAIs), defined as infections acquired by patients during hospitalisation or following medical procedures in healthcare facilities, are a persistent global challenge, causing significant morbidity, mortality, and increased healthcare costs [1,2]. The COVID-19 pandemic has had a significant impact on healthcare systems, highlighting their deficiencies and vulnerabilities, with a massive impact on hospital-acquired infections [3]. Data from the European Centre for Disease Prevention and Control (ECDC) showed that during the pandemic peaks, there was a significant increase in infections related to medical equipment (CLABSI, CAUTI, ventilator-associated infections, MRSA bacteraemia) [4,5]. At the same time, the reinforcement of infection prevention and control (IPC) measures, such as improved hand hygiene and the correct use of personal protective equipment, contributes to a reduction in specific categories of infection [4].

Although extensive literature exists on HAIs at local and national levels, there is a lack of comprehensive global or cross-continental analyses comparing IPC practices during the COVID-19 pandemic, and systematic data from low- and middle-income regions remain limited [5]. Identifying these gaps is crucial, as recognising regional differences and weaknesses in IPC strategies allows health systems to take targeted action, reduce infectious diseases, limit antimicrobial resistance, and develop sustainable health policies [1]. The lessons learned during the pandemic provide important insights for improving IPC programmes and developing strategies for future health crises.

The study aims to systematically review the global literature on healthcare-associated infections (HAIs) during the COVID-19 pandemic, identify regional best practices for prevention, and highlight areas requiring urgent improvements in IPC.

The COVID-19 pandemic-induced crisis has led to an increase in patients exhibiting severe conditions, placing an extraordinary strain on healthcare systems. Hospitals were required to modify their infrastructure by transforming specific departments or areas into temporary intensive care units. This strategy, while essential, placed significant strain on medical staff [6,7,8]. This situation significantly increased the risk of infection transmission between patients and medical staff and further jeopardised patient care, as facilities were not adequately prepared for the pandemic [6,7,8]. Consequently, patients admitted to intensive care units were found to be 20–28% more likely to develop a hospital-acquired infection compared to those treated in non-critical care units [9].

Rosenthal et al. (2022) [5] provide a comprehensive picture of HAI incidence in intensive care units in low- and middle-income countries in 2019 and the first five months of the pandemic, indicating that no similar approaches are currently available globally [5]. In Europe, it is estimated that more than 4.3 million hospitalised patients suffer from at least one hospital-acquired infection [10]. The WHO warns that healthcare-associated infections not only cause great suffering for patients but also result in high costs and increase antimicrobial resistance [1]. Understanding regional differences and adapting IPC strategies to the specific needs of individual healthcare systems can lead to more effective healthcare policies that reduce HAIs.

Important lessons for improving health policy can be learned from the various strategies employed to address healthcare-associated infections during the COVID-19 pandemic and its aftermath, as well as from understanding the gaps in healthcare systems. These findings support the identification of key factors and the development of recommendations for the sustainable management of healthcare-associated infections.

This systematic review integrates data from the global literature, highlighting regional best practices in preventing healthcare-associated infections during the COVID-19 pandemic. It also draws attention to improvements needed in IPC programmes.

## 2. Materials and Methods

This systematic review was conducted in accordance with the PRISMA (Preferred Reporting Items for Systematic Reviews and Meta-Analyses) guidelines. The review was prospectively registered in the PROSPERO database (International Prospective Register of Systematic Reviews) under the registration number CRD420251032525.

### 2.1. Search Strategy

To identify suitable studies, a comprehensive search was conducted in the following electronic databases: PubMed, Scopus, Web of Science using the following search formula: (“nosocomial infection” OR “hospital-acquired infection” OR “healthcare-associated infection” OR “cross infection” OR “HAIs”) AND (“COVID-19” OR “SARS-CoV-2”) AND (“infection control” OR “preventive measures” OR “disinfection strategies” OR “ surveillance systems”) AND (“post-pandemic strategies” OR “lessons learned” OR “pandemic response” OR “post-pandemic era”).

The research question guiding this review was structured according to the PICO (patient/problem, intervention, comparison and outcome) framework as follows: “How can region-specific infection control strategies (I) be compared with those of other regions (C) in healthcare facilities managing hospital-acquired infections during and after the COVID-19 pandemic (P) to reduce infection rates and improve patient and staff safety (O)?”.

### 2.2. Criteria for Eligibility

Studies that met the following criteria were considered for further evaluation: studies published in English between 1 January 2020, and 31 March 2025, and available as free full-text with a structured methodology. The following filters were applied in PubMed: free full text; human studies. Exclusion criteria were articles in languages other than English, book chapters, books, conference abstracts, editorials, expert opinions, and narrative reviews without structured methodology.

### 2.3. Data Extraction

The citations retrieved from PubMed, Scopus, and Web of Science were imported into R 4.5.0 for management and analysis. Duplicates were identified and removed, then selected by title and finally by abstract.

### 2.4. Assessment of the Risk of Bias

The risk of bias in the included studies was assessed using the ROBINS-1 V2 (Risk of Bias in Non-randomized Studies of Interventions, Version 2) tool, which was developed specifically for assessing non-randomised observational studies. This tool enables the assessment of material bias in ten key areas: Confounding Bias, Bias in Intervention Classification, Bias in Selection of Study Participants, Bias in Deviance, Bias in Missing Data, Bias in Outcome Measurement, and Bias in Selection of Reported Outcomes.

The assessment was conducted by two reviewers (C.V., A-C.G.), and all disagreements were resolved by consensus. This rigorous approach ensures a transparent and comprehensive assessment of the internal validity of all studies included in the review.

## 3. Results

### 3.1. Search

The search yielded 63 publications, and after removing 32 duplicates, 31 studies were identified for screening. After a subsequent screening of titles and abstracts, 14 studies were excluded because they did not meet the predefined inclusion criteria (e.g., incorrect study design, lack of structured methodology, or failure to address infection prevention and control in the context of COVID-19), and eight studies were selected for full-text review (Figure 1).

A total of 8 studies met the eligibility criteria and are listed in Table 1, which contains information on authorship, year of publication, study design, study period, sample, methodology, and key findings.

### 3.2. Characteristics of the Included Studies

The eight articles were published between 2020 and 2024. They came from different regions: Europe (UK, The Netherlands), Asia (Japan, China), North America (USA, Mexico), East Africa (Tanzania, Ethiopia, Kenya, Uganda), and one study with global coverage.

-The design was predominantly observational (*n* = 5), supplemented by a focus group study, a competency model, and an international survey.-Samples ranged from 7 patients in a single clinical trial to over 3200 healthcare workers in a multicentre programme.-The methodology included retrospective analyses, observational audits, focus groups, Delphi questionnaires, and online surveys.-The objectives focused on the investigation of nosocomial outbreaks, the experience of healthcare staff, the development of IPC skills, the implementation of programmes in resourcelimited areas, the design of hospital infrastructure, and the reporting of secondary bacterial outbreaks.

### 3.3. Risk of Bias Assessment

For each study, an overall assessment of risk of bias was made based on the highest level of bias found in a domain. The assessments were categorised as low, moderate, severe, or critical risk of bias (Figure 2).

Several important methodological limitations must be considered when interpreting this systematic overview (Figure 3).

Firstly, most of the included studies were observational studies conducted at a single centre and did not include randomisation or control groups. This carries a high risk of confounding, as differences in patient characteristics or local epidemiological factors may have influenced the results independently of the interventions [11,13]. The lack of large, diverse samples further limits the generalisability of conclusions to different populations and regions.

Secondly, many studies reported short follow-up periods, which hindered the evaluation of the long-term effects of IPC measures. Considerable heterogeneity in definitions and reporting of infections also compromised comparability, raising the possibility that differences in outcomes reflected methodological inconsistencies rather than actual variation in intervention effects [12].

The potential risk of bias is a significant concern. The application of the ROBINS-I tool revealed limitations in transparency due to inadequate reporting at the domain level. Significant biases were found in several studies, primarily due to the pooling of participants and the misclassification of interventions. The biases may have led to either underestimating or overestimating the actual effectiveness of IPC strategies. In addition, the prevalence of bias in outcome measurement and selective reporting has reduced confidence in the results presented. Consequently, the certainty of evidence is low, necessitating a cautious interpretation of conclusions [14].

Geographical limitations must also be considered. Most studies are from high-income regions, while data from resource-poor areas remain scarce. This imbalance may lead to selection bias and provide an incomplete picture of global differences in IPC strategies [16].

Finally, few studies examined the unintended negative consequences of IPC interventions, such as their burden on the healthcare workforce or the financial sustainability of health systems. The omission of these aspects limits a balanced evaluation of the overall impact of IPC measures [2].

Taken together, these limitations highlight the need for future multicentre, longitudinal, and methodologically rigorous studies. Only through higher-quality evidence can effective and generalizable global strategies be developed to prevent healthcare-associated infections.

## 4. Discussion

Several studies included in this review indicate that strict implementation of IPC measures contributed to reductions in healthcare-associated infections (HAIs). Conversely, the excessive burden on hospitals and diversion of resources for COVID-19 care have often disrupted routine IPC protocols and increased the incidence of HAIs [12,14,15,17]. Constant monitoring and ongoing training of healthcare staff were crucial to the success of IPC strategies. Facilities that established dedicated IPC teams and implemented intensive training programmes achieved a remarkable reduction in HAIs [18]. However, rapid restructuring of services—such as converting wards into COVID-19 units and redeploying staff—frequently undermined IPC practices and, in some cases, triggered hospital-acquired outbreaks [14].

Significant differences in the results of IPC were found between the different regions. In high-income countries, the availability of adequate resources, strong laboratory capacity, and robust surveillance systems facilitated the rapid and effective implementation of IPC measures, limiting the increase in HAIs [2,13]. Komasawa et al. in Japan highlighted the significance of early detection and containment [12]. In contrast, Koopsen et al. in the Netherlands illustrated that stringent IPC protocols effectively managed an Alpha variant outbreak despite the absence of COVID-specific infrastructure [13]. Conversely, nations with constrained resources faced challenges such as shortages of protective equipment, inadequate infrastructure, and a lack of sufficient personnel, leading to uncontrolled hospital-acquired transmission [11,19].

Progress was noted in Latin America and Eastern Europe, where international co-operation and the rapid adoption of IPC standards partially mitigated structural limitations [16]. A Spanish study emphasised that inadequate staffing and infrastructure hindered infection prevention and control during outbreaks [20]. Our findings are also supported by an Australian study, which demonstrated that nosocomial COVID-19 infections continue to occur during the vaccination period in vulnerable patient groups and in the absence of strict triage and isolation measures [21]. In contrast, Dave et al. found that in populations with high vaccination rates, such as during the Omicron wave, mortality was no longer directly linked to healthcare-associated infections, despite persistently high infection rates [22]. These findings suggest that success depended not only on financial resources but also on the adaptability of health systems, vaccination coverage, and the ability to mobilise international support.

Socio-economic and infrastructural factors significantly influenced the results of the IPC. In high-income regions, continued investment in healthcare has ensured sufficient critical care capacity, stable supply chains, and reliable access to personal protective equipment [2,13]. Digital health infrastructure, with electronic reporting systems and telemedicine, facilitated real-time surveillance and improved outbreak response [18].

In low- and middle-income countries, chronic underfunding has limited the availability of diagnostic facilities, trained personnel, and protective equipment [11,16]. The urban–rural divide exacerbated these challenges. Urban tertiary hospitals sometimes received international aid, while rural facilities often lacked basic infection control infrastructure. In addition, the migration of labour to wealthier areas exacerbated the shortage of skilled professionals, reducing local IPC capacity [11,16]. The inequalities led to different epidemiological outcomes: In Japan and Western Europe, outbreaks were effectively contained through rigorous testing and isolation measures [11,19]. At the same time, certain regions in Latin America and Sub-Saharan Africa experienced inconsistent implementation of infection prevention and control [16]. In Eastern Europe, constrained spending and workforce limitations were somewhat alleviated by the rapid implementation of EU and WHO IPC standards [16], underscoring the significance of policy alignment and international collaboration in addressing socio-economic challenges.

Although, the work of Assi et al. is not region-specific, it emphasises the potential of integrating antimicrobial stewardship, infection prevention, and diagnostic stewardship into a unified framework, which is particularly relevant for resource-limited facilities where efficiency and infrastructure sharing are critical [18]. Rangel et al. illustrate how the global spread of extensively drug-resistant Acinetobacter baumannii during the pandemic magnified IPC challenges in both high- and low-income regions [23]. These findings underline that antimicrobial resistance (AMR) is a transregional challenge that intersects with IPC and requires integrated strategies worldwide.

The overall certainty of the evidence remains low, as most included studies were observational, single-centre analyses with short follow-up periods [12,14,15]. Under-reporting, particularly in rural and low-resource regions, further limits generalisability [16,20]. According to Cochrane guidance, these methodological limitations and risks of bias reduce confidence in the observed associations and call for cautious interpretation [12,14,15].

Based on the available evidence, several priorities emerge for strengthening global IPC strategies:Develop and regularly update national and international IPC guidelines to minimise regional disparities [2,13,16].Strengthen infrastructure and ensure equitable resource distribution, with special focus on rural and low-resource settings [11,16].Implement continuous training and psychosocial support for healthcare workers to reduce burnout and sustain IPC compliance [15,18].Establish integrated surveillance platforms for HAIs and AMR, providing real-time feedback to institutions [18,23].Foster international cooperation to facilitate knowledge transfer, resource mobilisation, and harmonisation of IPC standards [16].

## 5. Limitations

This study presents several limitations that warrant acknowledgement.

Geographic imbalance: The majority of studies were conducted in high-income countries, with minimal data available from low- and middle-income settings. This limits the external validity and generalisability of the results.

Small number of studies per category: Each thematic area was represented by only one or two studies, which reduces the certainty of the evidence.

Heterogeneity: The included studies vary substantially in terms of design, population, healthcare settings, and interventions. This heterogeneity prevented pooling of data and precluded a meta-analysis.

Publication bias: There is a potential risk of publication bias, as outbreaks that are large, severe, or unusual are more likely to be reported, possibly skewing the available evidence.

Narrative synthesis only: Due to methodological and contextual variability, the findings are based on narrative synthesis rather than quantitative analysis, which increases subjectivity in interpretation.

Temporal limitation: The majority of studies were carried out during the first pandemic waves (2020–2022). The evidence may fail to reflect subsequent improvements in infection prevention and control practices.

## 6. Conclusions

This systematic review shows that the COVID-19 pandemic has significantly impacted infection prevention and control programmes, but also provides valuable lessons for strengthening the resilience of health systems. Countries with sufficient resources were able to reduce the number of infectious diseases by rapidly implementing surveillance measures, training staff, and adapting infrastructure. In contrast, regions with limited resources faced an increase in infections but benefited from international cooperation and the transfer of best practices.

Looking ahead, effective HAI prevention requires flexible IPC standards, sustained investment in infrastructure and human resources, the integration of antimicrobial stewardship, and the development of international collaborative mechanisms. Multicentre and long-term research is needed to assess the long-term impact of interventions and to provide generalisable evidence, especially in low- and middle-income countries. Strengthening equity in health investments remains a global priority.

## Figures and Tables

**Figure 1 jcm-14-06654-f001:**
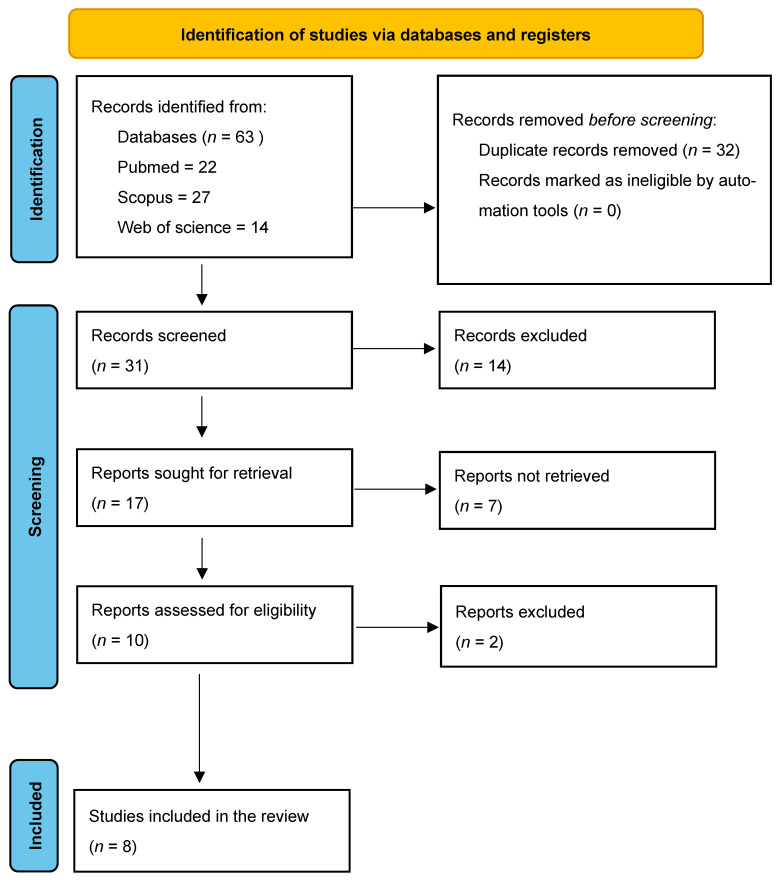
Our customised PRISMA flowchart outlines the process of study identification, screening, eligibility, and inclusion.

**Figure 2 jcm-14-06654-f002:**
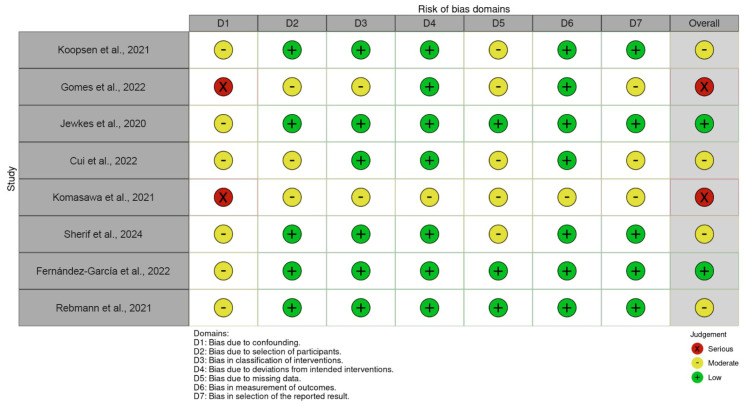
Risk of bias in the individual studies [2,11,12,13,14,15,16,17].

**Figure 3 jcm-14-06654-f003:**
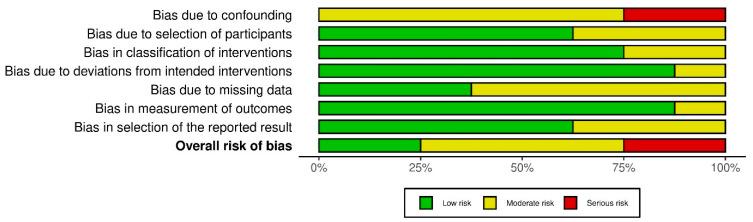
Risk of bias assessment across domains.

**Table 1 jcm-14-06654-t001:** Characteristics and main features of studies investigating healthcare-associated infections during and after the COVID-19 pandemic.

Author (Year)	Country/Region	Study Design	Study Period	Sample	Methodology	Key Findings
Jewkes et al. (2020) [11]	United Kingdom	Observational study	2020	Neurology department	Retrospective analysis	Outbreak containment, transparency, and preventive measures are essential
Komasawa et al. (2021) [12]	Japan	Multicentre observational study	2020–2021	3 hospitals	Observational, internal surveys	Early detection, rapid PCR testing, and adaptability are required for outbreak control
Koopsen et al. (2021) [13]	Netherlands	Observational audit	2020	11 nurses	Genomic surveillance	Importance of reinfection detection (83 days)
Rebmann et al. (2021) [2]	USA	Focus groups	2020	7 focus groups	Thematic focus groups	Reported PPE shortages, staff burnout, limited testing, and psychological stress
Cui et al. (2022) [14]	China	Competency model development	2020–2021	Model validated with IPC practitioners	Delphi method, questionnaires	Model with five domains (skills, management, collaboration, crisis response, IT)
Fernández-García et al. (2022) [15]	Mexico	Observational study	2020	7 patients	Clinical analysis	Risk factors: overcrowding, PPE misuse, antibiotic misuse; outbreak controlled with reinforced IPC
Gomes et al. (2022) [16]	East Africa (Tanzania, Ethiopia, Kenya, Uganda)	Multicentre IPC programme	2020–2022	>3200 healthcare workers	Mentorship and training programme	Improved IPC capacity; long-term systemic strengthening; Ebola experience leveraged in Uganda
Sherif & Iskandar (2024) [17]	Global (7 regions)	Survey study	2022	56 medical design professionals	Online survey	Consensus on single ICU rooms, circulation separation, hand-washing stations, antimicrobial surfaces, and non-contact tech

## Data Availability

All data generated or analysed during this study are included in this published article.

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
