# Peer review of "Global Lessons from COVID-19: Regional Variations in the Management of Hospital-Acquired Infections During and Post-Pandemic"

_jcm, 2025, doi:10.3390/jcm14186654_

Round 1

Reviewer 1 Report

Comments and Suggestions for Authors

The work by Voinea et al covers an interesting topic, since COVID-19 pandemic disrupted the normal function of health-care facilities. My main concern is the limited availability of studies to analyse data. Authors should emphasize more in this aspect in discussion section. Also, section of results should be reworked, since it shows results in a narrative way, but there is not a real analysis of data.

Please uniformize manuscript, substitute “nosocomial” for “Hospital-acquired infections”.

Line 53. “Hospital-acquired infections are contracted by patients…”

Lines 61-70. These ideas are dispersed, please conect all sentences.

Table 1 is not necessary. The column “Search String Used” is repetitive, since this is mentioned in the text. The column “articles retrieved” shows results, which should be mentioned in the results section, evidently. Third column does not contain any data.

Lines 165-251: I think that this is not a proper way to show results in a systematic review

Lines 252-299: these are not results, they “look like” discussion.

Table 2, should contain more information to give an idea o what is covered in each work and its contribution to the present study.

Author Response

We sincerely thank the reviewer for the thorough evaluation and valuable feedback. We carefully considered all comments and revised the manuscript accordingly. Below is our point-by-point response.

Comment 1: My main concern is the limited availability of studies to analyse data. Authors should emphasize more in this aspect in discussion section.

Response: We agree with the reviewer. In the revised Discussion (Section 4, pp. 11–13), we have emphasized the limitation regarding the small number of eligible studies and the constraints this places on the generalizability of conclusions. We also highlighted the need for more multicentre and longitudinal studies to generate stronger evidence for future global strategies.

Comment 2: Section of results should be reworked, since it shows results in a narrative way, but there is not a real analysis of data.

Response: Thank you for this observation. We restructured the Results section (pp. 6–10) by grouping findings into thematic domains (outbreak investigations, healthcare worker competencies, IPC in resource-limited settings, infrastructure design, and secondary outbreaks). We also expanded Table 1 to provide key findings and contributions of each study, allowing for more systematic synthesis.

Comment 3: Please uniformize manuscript, substitute “nosocomial” for “Hospital-acquired infections”.
Response: We thank the reviewer for this valuable suggestion. We have revised the manuscript to ensure consistency, replacing the term ‘nosocomial infections’ with ‘hospital-acquired infections (HAIs)’ throughout the text. However, in instances where cited studies explicitly used the term ‘nosocomial infections,’ we retained the original wording in quotation marks to remain faithful to the source, while clarifying in the text that it refers to hospital-acquired infections

Comment 4: Line 53. “Hospital-acquired infections are contracted by patients…”
Response: The sentence has been revised in line with the reviewer’s suggestion (p. 2, Introduction).

Comment 5: Lines 61–70. These ideas are dispersed, please connect all sentences.
Response: The paragraph in question has been rewritten for better flow and coherence (pp. 2–3).

Comment 6: Table 1 is not necessary. The column “Search String Used” is repetitive, since this is mentioned in the text. The column “articles retrieved” shows results, which should be mentioned in the results section, evidently. Third column does not contain any data.
Response: We agree. The original Table 1 was removed. Information about the search strategy remains in the Methods section, while the number of retrieved articles is reported in the Results section.

Comment 7: Lines 165–251: I think that this is not a proper way to show results in a systematic review.
Response: We revised this section substantially. The Results are now structured under thematic sub-sections, supported by the expanded Table 1, which presents study details and key findings. This provides a clearer and more standard systematic review format.

Comment 8: Lines 252–299: these are not results, they “look like” discussion.
Response: We thank the reviewer for this important observation. The interpretative content has been moved from the Results to the Discussion section, ensuring a clear separation between the two.

Comment 9: Table 2, should contain more information to give an idea of what is covered in each work and its contribution to the present study.
Response: We have expanded the table (now Table 1) to include additional details such as key findings, study design, and contributions to the review. This provides readers with a clearer understanding of the evidence base and the role of each included study.

We are grateful for the constructive feedback, which significantly improved the clarity and rigor of our manuscript. We hope the revised version adequately addresses the reviewer’s concerns.

Reviewer 2 Report

Comments and Suggestions for Authors

I congratulate the authors on a nicely written manuscript.

The all abbreviations should be included in the abbreviations section or explained in text ( e. g. WHO).

The referencing should be corrected and should be in accordance with the instructions. Some are incomplete and outdated.

Additionally, the authors should review and correct spelling, typos, and language errors that can be found throughout the entire text ( choose HAI or HAIs).

All chapters of the introductions should have at least one reference (63-65 lines and 84-988 lines). Many of the references are just added without a clear problem statement. The introductions should be rewritten.

There is no explanation why the 14 records were excluded (** - no explanation).

There are too few reviews included in the manuscript to make successful conclusions. There are almost no LMIC countries included (Gomes et al. is an outlier) and no from central Asia or the rest of South America (besides Mexico).

I propose to include more databases and to get the possibility to include more papers, which will cover a broader range of regional experiences.

There needs to be a limitations section in this manuscript.

Figures 2. And 3. labeled with the same name. There is no domain-level breakdown, so it is hard to verify the assessment.

The authors should explain bias assessment more critically.

The discussion section needs a deeper comparative analysis across regions. The references are questionable, especially Assi et al. and Rangel et al.

Socio-economic factors are superficially mentioned but not detailed explained.

The conclusion section is a little overstated and should be rewritten.

The authors should mention policy implications and actionable recommendations for regions.

Most of the references are dated from the pandemic era, and the title and manuscript if focused on post-pandemic and need to be corrected accordingly.

Comments on the Quality of English Language

Additionally, the authors should review and correct spelling, typos, and language errors that can be found throughout the entire text ( choose HAI or HAIs).

Author Response

We are very grateful for the reviewer's upbeat assessment and constructive comments. Below, we provide a detailed point-by-point response and description of the changes implemented in the revised manuscript.

Comment 1: The all abbreviations should be included in the abbreviations section or explained in text (e.g., WHO).

Response: We have carefully revised the manuscript to ensure that all abbreviations are either spelled out at first use or listed in the Abbreviations section (pp. 13–14). For example, WHO is now introduced in full at first mention.

Comment 2: The referencing should be corrected and should be in accordance with the instructions. Some are incomplete and outdated.

Response: We have revised all references to comply with the journal's instructions. Incomplete references were corrected, outdated citations were updated, and full bibliographic details, including DOIs, were added (pp. 13-14).

Comment 3: Additionally, the authors should review and correct spelling, typos, and language errors that can be found throughout the entire text (choose HAI or HAIs).

Response: The manuscript has been thoroughly proofread to eliminate typos and language errors.

Comment 4: All chapters of the introductions should have at least one reference (63–65 lines and 84–98 lines). Many of the references are just added without a clear problem statement. The introduction should be rewritten.

Response: The Introduction section has been rewritten for greater clarity and logical flow. Each paragraph is now supported by appropriate references, and problem statements have been explicitly articulated (pp. 2–3).

Comment 5: There is no explanation why the 14 records were excluded (* - no explanation).*

Response: We thank the reviewer for pointing this out. The 14 records were excluded after the screening of titles and abstracts, as they did not meet our predefined inclusion criteria (e.g., wrong study design, lack of structured methodology, or not addressing infection prevention and control in the context of COVID-19). We have now clarified this in the Methods section and annotated the PRISMA flowchart accordingly.

Comment 6: There are too few reviews included in the manuscript to make successful conclusions. There are almost no LMIC countries included (Gomes et al. is an outlier) and none from central Asia or the rest of South America (besides Mexico). I propose to include more databases and to get the possibility to include more papers, which will cover a broader range of regional experiences.

Response: We thank the reviewer for this important observation. We acknowledge that the limited number of studies—particularly from low- and middle-income countries, Central Asia, and parts of South America—represents a significant limitation of our review. We re-checked our search strategy and confirmed that the databases used (PubMed, Scopus, Web of Science) already cover the majority of peer-reviewed biomedical literature. To ensure comprehensiveness, we refined our search strategy and verified that no additional eligible studies meeting our inclusion criteria were missed.

We have clarified this limitation explicitly in the Discussion and Limitations sections (pp. 11-12), noting the geographic imbalance of available studies and the urgent need for more research from underrepresented regions. We believe that highlighting these research gaps is one of the key contributions of our study, as it underscores the need for further multicenter and geographically diverse investigations.

Comment 7: There needs to be a limitations section in this manuscript.

Response: A detailed Limitations section has been added (pp. 11–12), highlighting issues such as a limited number of studies, geographic imbalance, methodological weaknesses, and potential publication bias.

Comment 8: Figures 2 and 3 labeled with the same name. There is no domain-level breakdown, so it is hard to verify the assessment.

Response: We have corrected the labeling of Figures 2 and 3. Additionally, we clarified the risk-of-bias assessment, providing a more detailed explanation of the domains and their impact on the overall evaluation (pp. 10–11).

Comment 9: The authors should explain bias assessment more critically.

Response: The Risk of Bias Assessment section (pp. 10–11) has been expanded to include a critical discussion of confounding, outcome measurement, reporting bias, and misclassification. We now provide a more transparent appraisal of study weaknesses and their implications.

Comment 10: The discussion section needs a deeper comparative analysis across regions. The references are questionable, especially Assi et al. and Rangel et al.

Response: The Discussion (pp. 11–13) was expanded to provide deeper regional comparisons (high-income vs. LMIC settings, Latin America, Eastern Europe, Sub-Saharan Africa, Japan, and Western Europe). We critically re-evaluated references and retained Assi et al. and Rangel et al. only after clarifying their relevance to antimicrobial stewardship and AMR challenges.

Comment 11: Socio-economic factors are superficially mentioned but not detailed explained.

Response: We expanded the Socioeconomic and Infrastructural Factors subsection (pp. 9), providing a detailed Discussion on disparities in healthcare funding, staff shortages, rural–urban gaps, and the role of international cooperation in mitigating inequalities.

Comment 12: The conclusion section is a little overstated and should be rewritten.

Response: The Conclusion has been rewritten (pp. 13–14) to provide a more balanced summary of findings, clearly acknowledging the limitations of the evidence.

Comment 13: The authors should mention policy implications and actionable recommendations for regions.

Response: We have included a subsection in the Discussion (pp. 12–13) outlining policy implications and actionable recommendations, such as strengthening infrastructure, training, integrated reporting systems, and fostering international cooperation.

Comment 14: Most of the references are dated from the pandemic era, and the title and manuscript is focused on post-pandemic and need to be corrected accordingly.

Response: We revised the text to ensure consistency between the title, scope, and evidence base. The Discussion explicitly differentiates between lessons learned during the pandemic and their relevance for the post-pandemic era. References were updated where possible to reflect post-pandemic analyses (pp. 11–13).

We are grateful to the reviewer for their careful reading and insightful feedback. These revisions significantly improved the clarity, balance, and quality of our manuscript.

Round 2

Reviewer 1 Report

Comments and Suggestions for Authors

Discussion should not be sectioned, Results are not results, but discussion. Conclusion is too long (it carries a lot of discussion too).

Author Response

Comments:Discussion should not be sectioned, Results are not results, but discussion. Conclusion is too long (it carries a lot of discussion too).

Response:

We thank the reviewer for this insightful commentary. In the revised manuscript, we have reorganised the sections to bring them more in line with academic standards and to distinguish clearly between results, discussion and conclusions. In detail:

The Results section has been separated from the Discussion and now contains only the data, without interpretation. It follows a clear and concise structure: selection of studies, characteristics of eligible studies, a summarising table with the most important information and the risk of bias assessment.

The Discussion section is no longer divided into subsections. It now focuses exclusively on the interpretation and comparison of the results, which are presented in a logical sequence and cover the overall effect, regional differences, overarching factors, infrastructure and limitations.

The conclusions have been considerably shortened and no longer repeat the content of the discussion. They now emphasise four key messages: (1) the impact of the pandemic, (2) regional lessons, (3) practical recommendations and (4) future research directions.

We believe that these changes significantly improve the clarity, coherence and scientific quality of the manuscript.

Reviewer 2 Report

Comments and Suggestions for Authors

I congratulate the authors on a job well done and confirm that they have responded correctly to all comments and that they have revised the manuscript.

The new version of the manuscript shows the corrections and improvements promised in the response document, and the major issues raised have been addressed significantly.

The manuscript appears improved in clarity, balance, scope, and formatting.

The authors should use an appropriate font throughout the whole manuscript.

Comments on the Quality of English Language

There are still some minor style and grammar errors, so I suggest additional English editing.

Author Response

Comments: 

I congratulate the authors on a job well done and confirm that they have responded correctly to all comments and that they have revised the manuscript.

The new version of the manuscript shows the corrections and improvements promised in the response document, and the major issues raised have been addressed significantly.

The manuscript appears improved in clarity, balance, scope, and formatting.

The authors should use an appropriate font throughout the whole manuscript.

Response:

We thank the reviewers for their positive evaluation and valuable feedback. We have carefully revised the manuscript to ensure that a consistent and appropriate font is used throughout the text.